# Fungal community and taxa specialization to host and environment interactions in two temperate forests

**Maria Soledad Benitez Ponce**[1]*, **Michelle H. Hersh**[2], **Lindsey Becker**[3,4],
**Rytas Vilgalys**[3], **James S. Clark**[3,5]

**1** Department of Plant Pathology, College of Food Agricultural and Environmental Sciences Wooster Campus, The Ohio State University, Wooster, Ohio, United States of America, **2** Department of Biology, Sarah Lawrence College, Bronxville, New York, United States of America, **3** Department of Biology, Duke University, Durham, North Carolina, United States of America, **4** Department of Entomology and Plant Pathology, North Carolina State University, Raleigh, North Carolina, United States of America, **5** Nicholas School of the Environment, Duke University, Durham, North Carolina, United States of America

* benitezponce.1@osu.edu (MSBP)

## Abstract

The structure and function of plant-associated fungal communities (i.e. mycobiome) is shaped by biotic and abiotic factors, and can impact plant community dynamics. We evaluated the effects of different environmental factors in structuring the communities of seedling-associated fungi in temperate tree species, considering both the Janzen-Connell hypothesis as well as the impacts of climate warming. We tested the hypothesis that fungal host-specialization is observed at both the individual fungus and fungal community levels and is modulated by environmental conditions. The seedling fungal communities were characterized from tree species grown in two forests, under experimental manipulation of light, warming, and distance to and density of conspecifics. Fungal communities were analyzed using generalized joint attribute models. While warming, light, and forest site played a role in structuring seedling fungal communities, host, distance to, and density of conspecifics were stronger contributors. Furthermore, we could identify which fungal taxa responded to which predictors. This work supports the concept of fungal host-specialization at the community level, and points to particular fungal taxa which may play roles in density- and distance-dependent regulation of plant species diversity in the studied forests.

## Introduction

That pathogens play a key role in regulating plant diversity via conspecific negative density dependence (CNDD) is now well established [1,2]. Determining which pathogens are involved in this mechanism has become more of a challenge than initially anticipated. Early studies documenting distance- and density-dependent seedling survival assumed that these patterns were driven by host-specific natural enemies

**Data availability statement:** All sequences and associated metadata have been deposited in Qiita (https://qiita.ucsd.edu/) and NCBI's Sequence Read Archive (https://www.ncbi.nlm.nih.gov/sra) under study ID 12978 and BioProject PRJNA1245455, respfectively. All other relevant data is included in the manuscript text and/or its Supporting Information files.

**Funding:** This work was funded by the National Science Foundation grant NSF-DEB-0955904 to Rytas Vilgalys and Jim Clark, co-authors of the manuscript. The funders of this research had no role in study design, data collection and analysis, decision to publish, or preparation of the manuscript.

**Competing interests:** The authors have declared that no competing interests exist.

[3,4]. But, as understanding of seedling fungal communities has deepened, this idea of strict host specificity has been recognized as an oversimplification, with many root-associated fungi colonizing multiple hosts and demonstrating different impacts on seedling survival based on host identity [5] or environmental context [6]. Further, seedling root mycobiomes are now recognized as quite complex, with many co-occurring fungal species [7], such that assessing the role of each individual fungal species is challenging. Finally, all of these interactions take place in the context of a warming climate; warming can increase the relative abundance of soil pathogens [8,9] and the strength of CNDD [9]. Here, we demonstrate an analytical approach that can identify specific fungal taxa responding to different predictors, including conspecific host density and distance as well as experimental warming, that may be critical in understanding distance and density-dependent responses in light of changing environmental context.

Patterns of plant abundance and demographic rates consistent with regulation by natural enemies [e.g. 10,11] have typically been discussed in the framework of the Janzen-Connell (JC) hypothesis [3,4]. Under this framework, plant diversity is promoted by the host-specific attack from plant pests and pathogens. Seedling recruitment is limited by host-specific enemies in areas where the host is locally abundant, thereby increasing diversity. Demographic evidence shows that high host density and/or short distance from conspecific adults decreases seedling survival and recruitment rates [11–13]. Diversity is maintained because ostensibly host-specific pathogens attack one species and not others, and are thereby able to act as a biotic filter that limits recruitment of individuals of the same host.

However, a growing body of evidence does not find strict host specificity in individual fungal pathogens, but instead effective specialization [14]. The requirement for host-specific pathogens [15] conflicts with observations that many of the individual fungal taxa associated with seedlings roots and stems are typically found in multiple hosts [6,16–19], but see also [20], though overall fungal community composition may differ between host species [21,22]. Negative interactions between pathogens and their tree hosts could be modulated by additional interactions with beneficial mycorrhizal fungi, especially ectomycorrhizae [23–25]. Further, environmental characteristics such as light [26,27] and water availability [17,28,29] could alter the nature of plant-pathogen interactions by altering fungal community structure, function, or both. These same environmental variables that alter fungal communities also directly affect plant health and plant community composition. For instance, seedling survival is differentially affected by abiotic conditions, such as light availability [30,31] and soil moisture [6], potentially contributing to niche differentiation in plants [30]. Under the warmer temperatures predicted under climate change, forests may experience a greater relative abundance of soil pathogens [8,9], higher foliar disease severity [32,33] or intensification of negative feedbacks [9,32]. However, other studies have found weaker feedbacks [34,35] and decreased pathogen abundance [34] under warmer temperatures. As climate warming progresses, an understanding of plant-soil feedbacks under warming becomes more critical, yet studies exploring these feedbacks under experimental warming treatments are sparse [36].

Benítez et al. [14] hypothesized that JC effects could result from effective specialization, where pathogen specialization to a host results from the interaction of the host and pathogen with particular local conditions. If considering multiple fungal species infecting one host, effective specialization will occur as a result of the combined effect of different pathogen infections in a host. These pathogen effects need not to be additive, and may each in turn be modulated by the environmental conditions (see also [6]). Therefore, different combinations of fungal infections will affect host species in different ways in a context-dependent manner.

Resolving the apparent contradictions between demographic studies that support JC and the lack of observed pathogen specificity, which does not support JC, requires analysis of plant-associated fungal communities and their joint relationships with host and environment. In this study, we test the hypothesis that fungal host specialization to tree species is observed for both individual fungal taxa and for communities of fungi, and that host specialization is modulated by environmental conditions. Seedling-associated fungal communities were intensively surveyed from experimental sites at two temperate forests, with an experimental design that incorporates treatments of light availability and elevated temperature as well as distance to and density of conspecifics. The seedling fungal communities (mycobiome) were characterized using high-throughput sequencing approaches. Generalized joint attribute modeling [GJAM; 37] was used to jointly predict both host health status and the host-associated mycobiome based on common predictors. Specifically, we tested a set of *a priori* hypotheses focused on biotic and abiotic factors which could influence seedling health status and the mycobiome of a given seedling and evaluated them using a model selection approach. We centered on distance from conspecific adults and conspecific seedling density as predictors relevant to the JC hypothesis, as well as environmental factors (light, site) which could modulate JC interactions. A subset of seedlings were exposed to an experimental warming treatment that allowed for prediction of changing fungal responses to elevated temperature.

## Materials and methods

### Field experiment design and seedling collection

To evaluate parameters impacting fungal communities and host specificity in forest seedlings, seedlings were planted and collected from experimental forest plots in the eastern USA. Two sites were studied at Duke Forest (North Carolina, USA) and Harvard Forest (Massachusetts, USA). Within each forest, sites differed in soil characteristics, topographic variation and dominant tree species. Additional details for each site is provided in S2A Table in S2 File, S1A Fig in S1 File, and S3 File (Supplementary Methods). At each site, two sets of replicated experimental plots were established. First, plots to study plant soil feedbacks (hereafter "PSF plots"), which consisted of 1-m$^2$ herbivore-exclusion cages located at known distances from conspecific adult trees, with far distance being > 10 meters, each with 80 planting positions. In addition, two levels of plant diversity were considered in PSF plots, monoculture (80 seeds of the same plant species) and high diversity (five species per plot, with a total of 80 seeds). Second, seeds of different tree species were planted in previously established experimental warming chambers, as described in [38] and S3 File, each with a maximum of 420 planting positions. The nine tree species included in this study are listed in S2B Table in S2 File, and were planted in PSF and/or warming plots, at each forest. Given climatic and compositional differences between DF and HF site, not all the same species were planted at each site. Both for the PSF plots and the warming chambers replicate treatments were installed in locations with varying light, corresponding to understory or opened gaps (see example in S1A Fig in S1 File), with a minimum of three PSF plots per distance or diversity treatment, per plant species established, and three chambers per warming treatment per site. Seeds were planted in each plot/site during late fall/early winter, prior to ground freezing, depending on each year's weather. First-year seedlings were monitored for seedling germination and survival on a weekly basis during the growing season (May-August) and sampled for fungal community characterization in 2011, 2012, and 2013 at all sites. Seedlings were sampled if they exhibited symptoms of pathogen, pest or abiotic damage (e.g. leaf damage and discoloration, chlorosis, wilting, stem damage/canker; collectively labeled as symptomatic seedling). Each time a symptomatic seedling was collected, an asymptomatic, conspecific neighbor, was also collected (if available within the plot). Collected seedlings were

kept cool (4–8 °C), transported to the laboratory, or shipped frozen (from HF to Duke), until processing. Seedlings were processed or frozen within 24–48 hours after sampling. The number of seedlings collected per species, per treatment and year of study differed based on germination rate, disease pressure and availability of asymptomatic conspecific neighbor. The number of seedlings included in this study are summarized in Table S2F in S2 File. The work performed at these sites followed Duke Forest and Harvard Forest guidelines for research registration and sampling.

### Fungal community characterization

In the laboratory, seedlings were washed in tap water and blotted on autoclaved paper towels. Stem and root tissue were processed separately and surface disinfested through a series of washes in 70% ethanol for 60 seconds, 1% sodium hypochlorite for 60 seconds, 70% ethanol for 30 seconds, followed by a sterile distilled water rinse. The tissue was then quickly frozen in liquid nitrogen and stored at -80 °C. Immediately prior to DNA extraction, seedling tissue was vacuum-dried and ground to fine powder using 4 mm sterile stainless-steel beads and 1-minute bead-beating in a Genogrinder 2000 (SpexSample Prep, Metuchen, NJ). Total DNA was extracted from ~100 mg ground tissue using either a CTAB protocol modified from [39,40] or with the NucleoSpin-96 Plant DNA extraction kit (Macherey-Nagel/ Clontech, Duren, Germany; with buffers PL2 and PL3). DNA extraction success and quality were confirmed through PCR amplification and gel electro-phoresis, prior to amplicon library preparation [41,42]. The seedling mycobiota was surveyed through amplicon sequencing of the large subunit of the ribosomal rDNA with primers LROR and LR3 (41,42; ~600 bp) in either 454-pyrosequencing (2011 samples) or Illumina MiSeq (2012 and 2013 samples) sequencing technologies. The transition from 454-pyrosequencing to Illumina became available to us only in 2012, and allowed us to increase the number of samples to be analyzed per sequencing run and maximize sequencing depth. Greater detail on amplicon library preparation is described in S3 File. All sequences and associated metadata have been deposited in Qiita [43; https://qiita.ucsd.edu/] and NCBI's Sequence Read Archive under study ID 12978 and BioProject PRJNA1245455, respectively.

For both 454 and Illumina data, sample de-multiplexing and initial quality filtering were performed using QIIME versions 1.7 and 1.8, respectively [44]. For 454 data, processing of demultiplexed files included denoising using Acacia [45] version 1.52, performed individually for each sequencing run. Denoised sequences were then merged and the UPARSE algorithm (USEARCH version 7) [46] was used for OTU calling at 97% similarity. For demultiplexed Illumina sequences, frame-shift primers were removed using cutadapt version 1.3 [47], followed by quality-based filtering and truncation, sequence de-replication, OTU-calling and generation of OTU-sample tables using USEARCH 7 [46] as described above. Illumina and 454- OTUs were compared with USEARCH 7 global alignment tool (usearch_global). Illumina OTUs were used as the database against which the 454-OTUs were queried to at a 99% similarity. Illumina and 454-OTU tables were then merged to include both OTU matches between platforms and OTUs unique to each. Taxonomy assignments of the final OTUs used in this research was based on comparison with SILVA's ribosomal RNA database LSU release from 3-30-2015 (https://www.arb-silva.de/), revised in 2022 and manually curated using BLAST. Any non-fungal/oomycete sequences were removed prior to downstream analysis. The final mycobiota dataset consisted of 2889 taxa (OTUs) recovered from 521 samples. The naming convention used in this manuscript for each OTU indicates the lowest taxonomic level of its classification (i.e. genus, family, order) followed by a number to differentiate between OTUs within the same taxonomic group. Data set statistics, including alpha and beta-diversity metrics were estimated in phyloseq package version 1.24.2 [48] in R [49] (S3 File). Effects of host species, symptomatic status, distance and density on diversity metrics were estimated using linear mixed effect models with year as a random effect. Analysis was separately run per forest, as different plant species were tested at each Duke and Harvard Forest. The total number of samples analyzed are summarized in S2F Table in S2 File.

### Models to determine fungal communities and seedling health responses to environmental variables

Fungal community and host responses to environmental variables were jointly modeled using Generalized Joint Attribute Modeling [37]. Generalized Joint Attribute Modeling (GJAM) is a multivariate Bayesian analysis which allows one to jointly

model different data types, in this case host health status (discrete and binary) and fungal OTUs (compositional data) measured with different sampling efforts (differences in sequencing depth per sequencing platform), while accommodating for the over representation of zeros, which is common in microbial community experiments [50,51]. A model selection approach was used to determine biotic and abiotic predictors of fungal community composition and host health status. Model combinations were built with variables known to affect seedling survival and fungal colonization in forest ecosystems, including components of distance and density dependent regulation of plant community composition (Table 1; [6,14]). In addition, preliminary tests were run with models that included methodological factors, specifically sequencing platform, sampling year, DNA extraction method, tissue type, and seed origin (S2C Table in S2 File), as these have shown to influence fungal community composition in other studies [52].

Models testing the effects of the biotic and abiotic predictors (host species, light availability; temperature; site; distance to adult tree and diversity of neighboring seedlings; and sequencing platform, S2D Table in S2 File) including the interactions between host and distance to conspecific adult (host*distance) and/or the interaction between host and diversity of neighboring seedlings (host*diversity) were run in R's *gjam* package version 2.5.2 [37]. For the analysis in *gjam*, seedlings from nine host species were included, corresponding to the nine species planted in the PSF plots, and for which host/density and host/diversity interactions could be analyzed (Table 1, S2B Table in S2 File). If seedlings from these nine species were planted and sampled from warming plots, they were also included in the analysis. Prior to the analysis, the fungal community data was filtered to include only fungal OTUs found in at least 10% of the studied samples, resulting in a total of 247 fungal OTUs from 521 samples. All models were run with *Acer saccharum* as the reference for the host variable and "low density" as reference for conspecific density. All models were run with Year as a covariate (S1D Fig in S1 File; S2D Table in S2 File). Models were evaluated based on their associated deviance information criterion (DIC) scores, and

**Table 1. Description of predictors considered in models of fungal community and host health responses.**

| Predictor | Description | Levels |
|---|---|---|
| Host | Species of seedling sampled for fungal community analysis | *A. saccharum* Marshall<br>*Carya alba* (Linnaeus)<br>*Fraxinus americana* L.<br>*Liquidambar styraciflua* L.<br>*Liriodendron tulipifera* L.<br>*Nyssa sylvatica* Marshall<br>*Pinus strobus* L.<br>*Q. falcata* Michx.<br>*Q. rubra* L. |
| Distance | Distance to closest conspecific adult | Far (>10 m)<br>Near (<10 m) |
| Density | Density of conspecific seedlings | High (all seedlings within a plot of the same species)<br>Low |
| Temperature | Refers to both air and soil temperature, if manipulated | Ambient (no manipulation)<br>Elevated (seedlings sampled within air and soil warming chambers) |
| Site | Six sampling sites were included in the analysis, with three sites each in Harvard Forest and Duke Forest. | Duke Forest Warming Site (DFWS)<br>Duke Forest Eno West (DFEW)<br>Duke Forest Blackwood (DFBW)<br>Harvard Forest Simes (HFST)<br>Harvard Forest Barre Woods (HFBW)<br>Harvard Forest Warming Site (HFWS) |
| Light | Light availability based on locations of open gaps at each site | Understory<br>Gap |
| Sequencing | Sequencing platform used for fungal community characterization | Roche 454 (R454)<br>Illumina MiSeq (MiSeq) |

the model with the lowest DIC score was chosen as the best model (S2D Table in S2 File) for further interpretation. The analysis with *gjam* resulted in the following outputs used for data interpretation: (A): Sensitivity analysis of response variables to individual predictors (and their interactions, Fig 1); (B): Predicted fungal OTUs and host health status responses to individual predictors (Figs 2, 3 and S1E Figs-S1I in S1 File), (C): Clustering of fungal OTUs and host health status based on similarity of responses to predictors (S1J Fig in S1 File).

## Results

### Data exploration and variable selection

The analysis of 521 seedlings resulted in a total of 5,485,558 sequences representing 2,889 LSU OTUs. On average, 13,039 and 816 sequences per sample were recovered from the Illumina MiSeq and 454-pyrosequencing runs, respectively, representing 126 (± 78) and 40 (± 26) OTUs per sample, per sequencing platform (S2E and S2F Tables). The taxa recovered, and included in the analysis, belonged to the Kingdoms Fungi and Oomycota. The ten most abundant OTUs (Colletotrichum_1, Diaporthales_3, Sordariomycetes_186, Amphisphaeriaceae_6, Ceratobasidium_2, Neofabraea_3, Cladosporium_1, Neofusicoccum_2, Chalara_1, Helotiales_26) were found in both symptomatic and asymptomatic seedlings of most host species analyzed (Fig S1B in S1 File). OTU richness (Chao1 index) recovered from seedlings at Duke Forest differed between host species and when recovered from seedlings at different densities of conspecific hosts (S2G Table, Fig S1C in S1 File). In addition, significant interactions between host species and distance to conspecifics, and interactions between distance to conspecific adults and density of conspecific seedlings were observed for Shannon diversity estimates from Duke Forest seedlings. For seedlings collected in Harvard Forest, host species had a significant effect on diversity estimates (Shannon index), and significant interactions were observed for OTU richness between symptomatic status and conspecific density (S2G Table in S2 File, S1C Fig in S1 File).

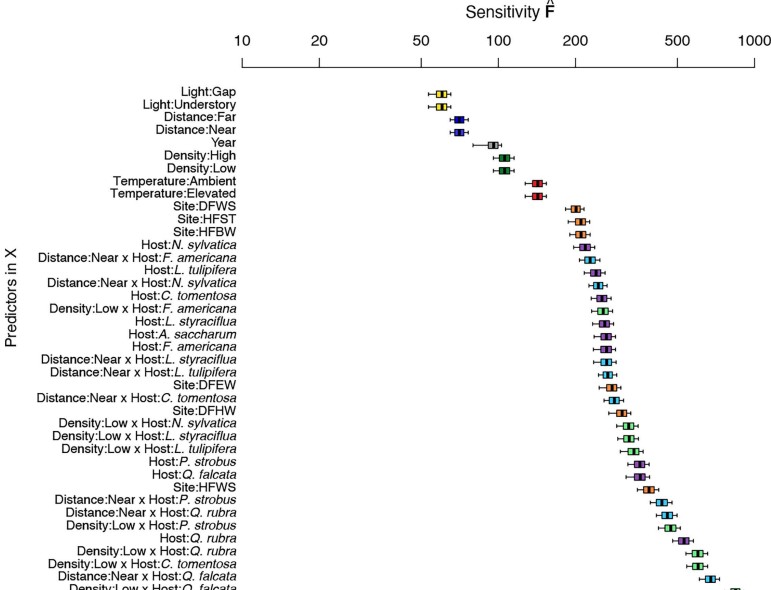

**Fig 1. Sensitivity of fungal OTUs and host health status to biotic (host, distance, and density), abiotic (temperature, site, light) and year as predictors.** Predictors are colored and labeled according to: Distance (blue) F = far, N = near; Density (green) HIGH = high density of conspecifics, LOW = low density of conspecifics; Light (yellow) G = gap, U = understory; TempTreat (red) A = ambient temperature, E = elevated temperature; Site (orange): DFEW, DFBW, DFWS = Duke Forest Eno West, Blackwood and Warming Site, respectively; HFBW, HFST, HFWS = Harvard Forest Barre Woods, Simes Tract and Warming Site, respectively; and hosts (purple, Table 1).

GJAM was applied to determine the strengths of nine methodological predictors on fungal communities and host health status. First, sensitivity analysis of these predictors indicated that year had the strongest impact on the recovered fungi (S1D Fig in S1 File), compared to other sampling and sample processing parameters; therefore, year was included as a covariate in subsequent models tested during the model selection approach. Other predictors (sequencing method, DNA extraction method) with a high sensitivity covaried with year, so we included only year in subsequent models. The model with the best fit, based on DIC scores, included the interactions between the predictors host and density of conspecifics (density * host), the interaction between the predictors host and distance to a conspecific adult (distance * host), and the abiotic predictors temperature, site, and light (S2D Table in S2 File).

## Community level responses to the model with the best fit

We used the best fitting model (S2D Table in S2 File) to explain the joint responses of fungal species and health status of the studied seedlings. The contributions of the individual predictors included in the selected model was compared via sensitivity analysis (Fig 1), where high sensitivity score indicates larger contribution of a predictor to the joint response of the 247 fungal taxa (recovered from the 521 seedlings) selected for analysis. Individually, forest site and host species were stronger predictors compared to other abiotic parameters (Fig 1). However, except for *F. americana*, the interaction between host species and density of conspecific seedlings and/or host species and distance to a conspecific adult was a stronger predictor of fungal community and host health than host identity alone (Fig 1).

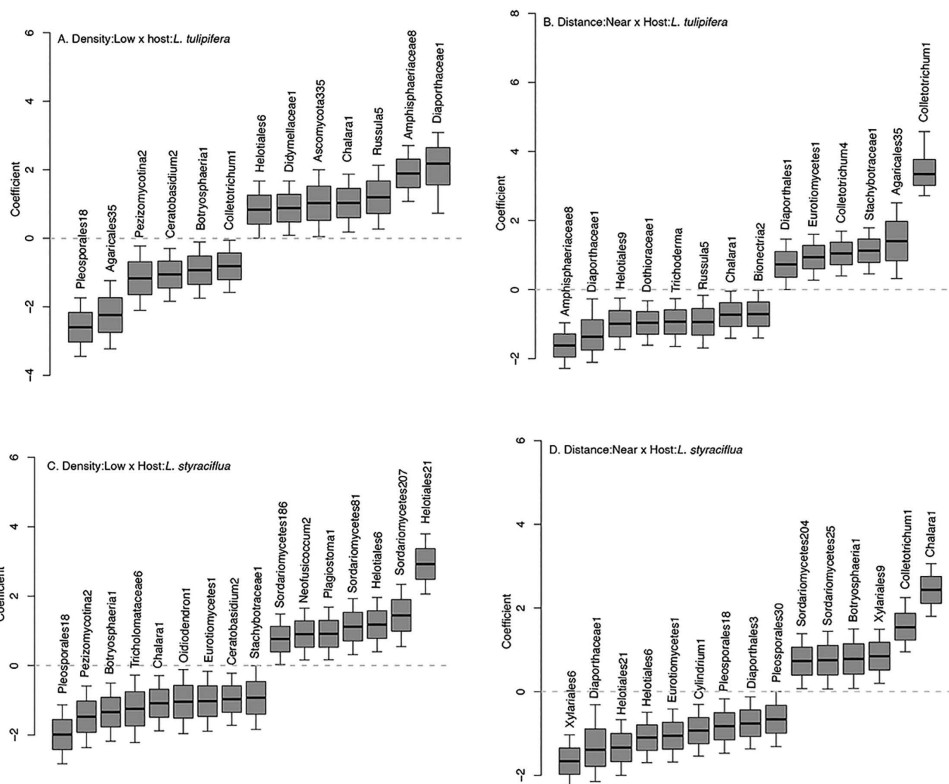

**Fig 2. Posterior distributions of fungal OTUs responses to host species *L.tulipifera* (A,B) and *L. styraciflua* (C,D) and their interaction with density of conspecific seedlings (A,C) or distance to a conspecific adult** (B,D). Taxa with credible intervals different than zero are shown. Results from other hosts and their interactions with density and distance are shown in Figs S1E, S1F and S1H in S1 File.

Interactions between host identity and density of conspecific seedlings (host*density), or host identity and distance to conspecifics (host*distance) also impacted the presence or absence of individual fungal taxa. GJAM evaluated response coefficients for all fungal taxa-predictor combinations, and Fig 2 summarizes individual fungal taxa responses to host*density and host*distance interactions for the plant hosts *L. styraciflua* and *L. tulipifera*. Results for other interactions are shown in S1E Figs and S1F in S1 File, and number of seedlings analyzed are summarized in S2F Table in S2 File. Individual fungal taxa were identified as preferentially recovered under each interaction when the response coefficient was greater than zero, or preferentially absent under each interaction when the response coefficient was less than zero. For instance, six fungal taxa were preferentially recovered (coefficient >0) from the host *L. tulipifera* when grown near a conspecific adult (Fig 2B). In addition, five of these taxa were different from the six fungal taxa preferentially recovered from the host *L. styraciflua* when grown near a conspecific adult (Fig 2D). Further, the fungus classified as Colletotrichum1 was preferentially recovered from both *L. styraciflua* and *L. tulipifera* when grown near conspecific adults, while the taxon Helotiales6 was preferentially recovered from both hosts when grown in areas of low density to conspecifics.

The number of individual fungal taxa responding to different predictors showed some patterns. Collectively, many fungal taxa responded (either positively or negatively) to the interactions between host and distance or host and density (Table 2, Figs S1E and S1F in S1 File, S2H Table in S2 File). More responses were observed for host*distance interactions (107) than host *density (90). In some cases, the number of fungal taxa responding to distance or density were asymmetric. For example, four times more fungal taxa responded to far distance from conspecific adults in *F. americana*,

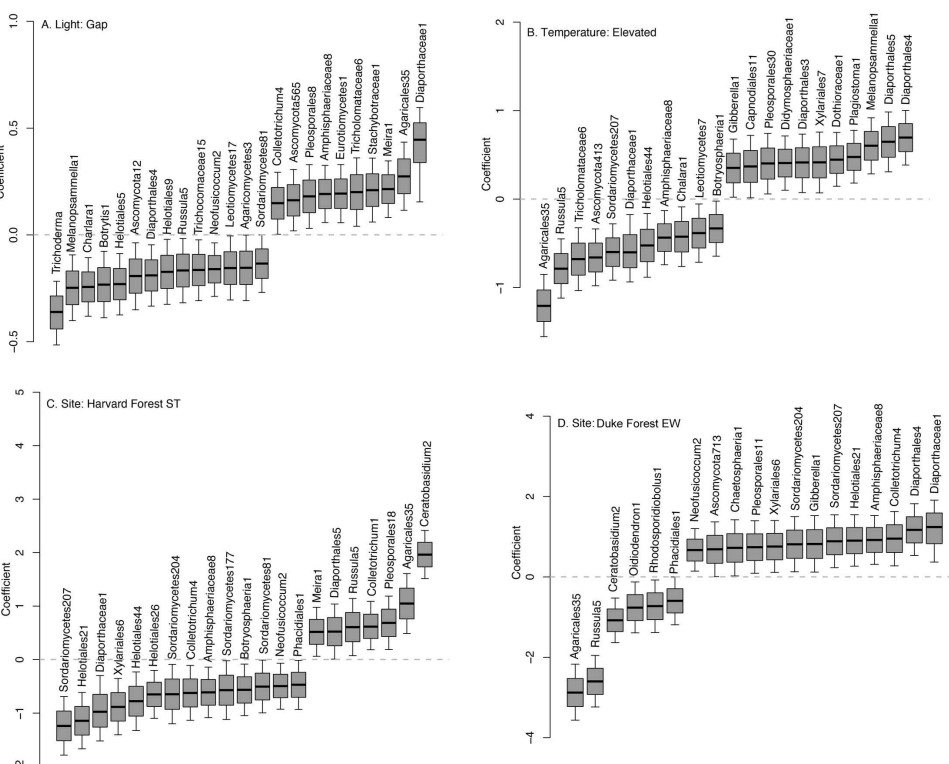

**Fig 3. Posterior distributions of fungal OTUs responses to environmental predictors, including light availability (A), elevated temperature (B), and site (C and D).** Taxa with credible intervals different than zero are shown. Light availability was controlled based on open gaps in the canopy (vs. understory). A subset of seedlings were exposed to elevated temperatures at Warming experiments at each forest (vs. ambient temperature). Two examples for site effect are shown for Harvard Forest Simes Tract (ST) and Duke Forest Eno West (EW). Additional sites are shown in Figure SIG in S1 File.

and five times more responded to far distance in *N. sylvatica.* Similarly, five times more taxa responded to low density when the host was *C. alba* and three times more taxa responded to high density when the host was *Q. rubra*.

Similar to host and conspecific density and distance interactions, different fungal taxa were preferentially recovered from seedlings sampled at different sites, and also within sites in locations that differed in light availability and warming treatments (Fig 3, Fig S1G in S1 File). The number of seedlings analyzed per site, and corresponding species are summarized in S2B Tables and S2F in S2 File. From the 247 fungal taxa included in the analysis, 38 and 25 taxa were preferentially recovered from Duke Forest and Harvard Forest sites, respectively. A subset of 22 or 24 taxa significantly responded to experimental warming and light availability, respectively. For example, one fungal taxon, Chalara1, was preferentially recovered from seedlings grown under low light (understory) and ambient temperature; whereas the taxon Melanopsammella1 was preferentially recovered from seedlings grown in the understory and under elevated temperature. Overall, eleven taxa were preferentially recovered under warmer temperatures, and eleven preferentially recovered under ambient (Fig 3B). Of the 22 taxa who responded to temperature, 18 also responded (either positively or negatively) to distance or density in at least one host species, indicating a potential interaction between temperature and distance or density. Site-specific patterns were also observed; for example, Colletotrichum1 was preferentially recovered from Harvard Forest seedlings, whereas Colletotrichum4 from Duke Forest seedlings.

Overall, from the 247 fungal taxa included in the analysis, 109 taxa showed preferential recovery when seedlings were grown under at least one of the studied predictors and/or their interactions (Fig 4), with a subset responding to multiple predictors. Six fungal taxa (Agaricales35, Russulaceae1, Colletotrichum1, Pleosporales18, Ceratobasidium2 and Diaporthaceae1) recovery were modeled at least half of the time (17 out of the possible 35 combinations; S2H Table). From these taxa, Colletotrichum1 and Diaporthaceae1 responded to eight and four of the host*distance combinations respectively, and Diaporthaceae1 to three of the host*density combinations. However, Colletotrichum1 was preferentially recovered from the various hosts at the near distance; whereas Diaporthaceae1 was preferentially recovered from three hosts at the distance farther from the conspecific adult (*L.styraciflua*, *L. tulipifera* and *N. sylvatica*) and one at closer distance (*Q. falcata*). Further, Diaporthaceae1 was recovered from seedlings of *F. americana*, *L. tulipifera* and *N. sylvatica* when grown under low conspecific density.

## Structural analysis of community level response

To determine if different taxa show similar patterns of responses to predictors, GJAM estimates the correlation between individual taxa responses to predictors and clusters taxa based on both these correlations and the posterior distributions of each taxa/predictor combinations. These taxa clustering is shown in the left panel of Figure SIJ in S1 File and the taxa clusters are summarized in S2I Table in S2 File. Analysis of the structure of responses of the fungal communities to

**Table 2. Number of fungal taxa responses to plant host identity, to the interactions between host identity with the density of conspecific seedlings, and to the interaction of host identity with distance to conspecific adults. Species abbreviation and additional species information are shown in S2B Table in S2 File.**

|  | acerSacc | caryTome | fraxAmer | liquStyr | liriTuli | nyssSylv | pinuStro | querFalc | querRubr |
|---|---|---|---|---|---|---|---|---|---|
| Host |  |  |  |  |  |  |  |  |  |
| Positive | 12 | 9 | 8 | 9 | 9 | 9 | 7 | 9 | 6 |
| Negative | 10 | 4 | 10 | 7 | 7 | 8 | 6 | 5 | 3 |
| Host*density |  |  |  |  |  |  |  |  |  |
| High | Ref | 1 | 9 | 9 | 6 | 6 | 6 | 4 | 3 |
| Low | Ref | 5 | 10 | 7 | 7 | 8 | 6 | 2 | 1 |
| Host*distance |  |  |  |  |  |  |  |  |  |
| Far | Ref | 6 | 12 | 9 | 8 | 20 | 7 | 2 | 6 |
| Near | Ref | 5 | 3 | 5 | 6 | 4 | 4 | 5 | 5 |

the tested predictors (Table 1) revealed four main clusters of taxa. Each cluster (labeled as 1–4 in Figure SIJ in S1 File) represents groups of taxa that responded similarly to the studied predictors. The clusters contained 139, 67, 12, and 29 fungal taxa, respectively. The response variable host health status was clustered together with the 139 taxa in group 1.

The impact of the predictors on the individual fungal taxa studied is shown in the right panel of Figure SIJ in S1 File. In this dataset, individual predictors clustered based on their contributions to the model (cluster shown in the top right panel of Figure SIJ in S1 File). The interaction between the host *Q. falcata* and low density of conspecifics separates from the rest of predictors. The rest of the predictors cluster based on distance and host species, density and host species interactions, or abiotic parameters. No consistent patterns are observed in the clustering of predictors among each taxa clusters (right panel, Figure SIJ in S1 File), indicating that while there is co-occurrence of certain taxa, it is not necessarily related to the predictors tested.

## Discussion

In this study, we tested the hypothesis that fungal communities in tree seedlings are modulated by host identity, conspecific seedling density, distance to conspecific adults, and several abiotic covariates, most notably experimental warming. We found that each tree species harbors a distinct community of fungi, but the structure of the fungal community is modulated by characteristics in ways that differ between host plants. Interactions between host identity and density of conspecifics, or host identity and distance to conspecific adults, exerted stronger influence over fungal communities than environmental variables that included warming, light availability, and forest site, and even host identity alone. Although fungal taxa are certainly generalists as gauged by the traditional view of numbers of hosts they infect, we found that the host by conspecific seedling density interaction and host by distance to conspecific adult interaction is fungal-taxon specific. This suggests that fungal host specialization occurred both at the individual taxon (OTU) level and the fungal community as a whole.

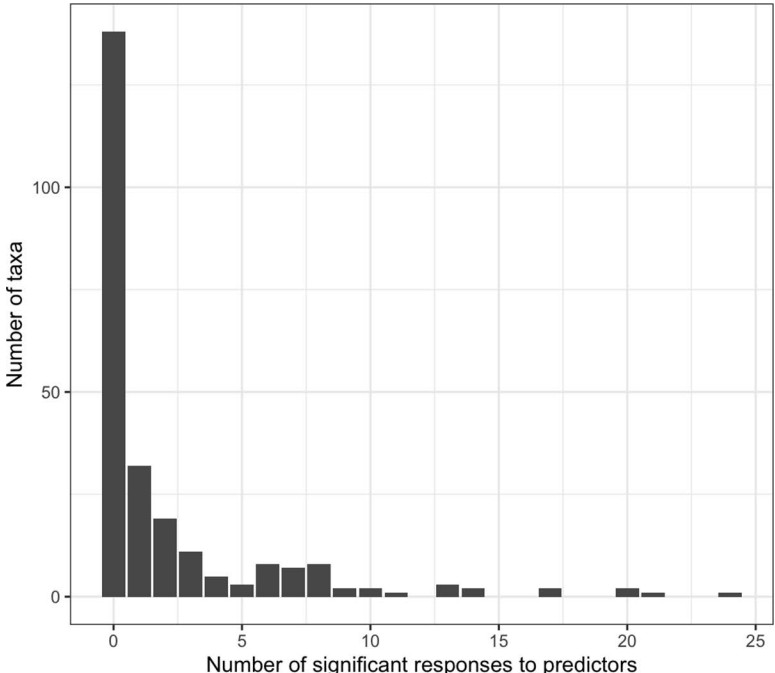

**Fig 4. Number of significant responses to predictors per taxa, based on posterior distributions being different from zero for the studied predictors and their combination.** The total number of responses per individual taxa is summarized, where the y-axis indicates total number of taxa per individual response number (x-axis).

Unique sets of fungal taxa responded to each biotic (i.e. distance to conspecific adults and density of conspecifics) and environmental (i.e. warming, light, site) variable. However, the responses of fungal OTUs to hosts depended on the presence and location of conspecific individuals. This result may explain previous observations of either distance or density dependent responses (but not both) in different forests [12,53,54]. Distance and density responses are not mutually exclusive, but instead both can lead to Janzen-Connell-like responses from different members of the fungal community [55]. For instance, most taxa responded to a limited number of predictors, but a small subset responded to more than ten.

Taxon-specific responses to host, site and environment (e.g. warming, light) were consistent with previous studies showing that seedling survival was differentially affected by combinations of generalist fungi, depending on soil moisture [6]. Host-associated microbial communities that are modulated by environment have been described in other plant species. For instance, host genotype effects of the herbaceous plant *Boechera stricta* on the leaf bacterial microbiome was modulated by site. Host genotype by site interactions could then be used to predict the abundance of bacterial taxa in *Boechera stricta* [56]. The set of fungal taxa that responded to different combinations of hosts and predictors set the stage for subsequent experimental studies that verify the role of different fungal taxa in seedling survival. Notably for future studies in these ecosystems, this work highlighted a subset of fungal taxa that responded, positively and negatively, to elevated temperatures consistent with a warming world. The majority of the species who responded to temperature (18/22, 81%) also had some kind of host-specific differential response to distance and/or density, indicating a potential interaction between the impacts of conspecific negative density dependence and warming that could be explored further. Large-scale experimental studies exploring the interactions between individual fungi and hosts [5] can be challenging to design given the many potential combinations; modeling approaches like this prune the list of taxa-host combinations that sets the stage for future experimental work. This is particularly important in studies of warming effects on plant-soil feedbacks, which are currently quite limited.

Communities of fungi did not respond equally to all predictors. Analysis of the structure of responses of taxa to predictors show clusters of fungi that differ in the strength to which they respond to the analyzed predictors. However, based on this analysis, it is not evident to which specific predictors these clusters are responding, or alternatively, the clustering is influenced by non-measured variables. In previous research [57–59], fungal communities are often treated uniformly, based on richness or evenness estimates or ordination axes as a dimension reduction strategy; or experimentally, through comparisons of comparing fungicide vs. non-fungicide treatment effect. Alternatively, specific host-pathogen interactions tend to be studied as a single interaction at a time. Instead, different members (or group of members) of a large and diverse community are responding differently to host and environmental factors collectively.

The application of GJAM models to the fungal sequencing survey presented in this work allowed an exhaustive analysis of the fungal communities to identify specialization of individual taxa and community. Individual taxa do not necessarily follow patterns of predicted functionality based on fungal taxonomy assignment and prior knowledge of fungal lifestyle, with the caveat that limited information on functionality is available given the taxonomy resolution used in this work. Changes in fungal community composition in response to conspecific density and distance to conspecific adults is observed for plant pathogens, beneficial symbionts and saprotrophs. While CNDD interactions have been reported previously for mycorrhizal host associations, we note that in this study fungal-host interactions were evaluated for seedlings > 1 year old, and that to our knowledge mycorrhizal CNDD interactions have not been noted for tree seedlings less than 1 year old [24,60]. For this reason, we did not evaluate CNDD within the context of host mycorrhizal status for this study. Further, the unequal germination between AM and EM tree species in this study confounded potential comparisons. However, hypotheses on shared functions could be developed for clusters of taxa with similar responses to predictors. For instance, subsets of the analyzed fungal OTUs share responses to the model predictors; and contrasting responses to predictors are observed by individual taxa. For example, though both Colletotrichum1 and Diaporthacea1 were recovered from several hosts and respond to multiple host and density interactions, Colletotrichum1 was preferentially recovered from distances near to a conspecific host; whereas Diaporthacea1 from further distances, as well as under low density of conspecifics. Determining

the functions of the recovered fungal taxa, however, remains dependent on better taxonomic resolution, along with additional functional data from methods such as enzyme assays, genomics, and pathogenicity analysis. Previous studies that had identified pathogen drivers of plant community dynamics focused on a combination of culture-based surveys [e.g., 6,16,61,62], amplicon based surveys [63–68] or fungicide disturbance treatments [69]. Even though cultured isolates provide an advantage when testing hypotheses of function, extensive surveys and factorial interactions combine to make it prohibitive to carry out these tests using diverse hosts and environmental conditions. Studies using a molecular approach have linked increased putative pathogen frequency to conspecific distance or density [19,21,67,70], but have not considered variation due to environmental factors.

By employing GJAM, this study was able to directly identify the responses of the fungal community and individual fungal OTUs to biotic and environmental variables, as well as interactions relevant to negative conspecific density dependence. It also highlights groups of fungi that share similar responses to predictors, and potentially functionality. The fact that different collections of fungi are associated with each tree, but that these fungi are responding to different suites of predictors, also suggests that effective specialization is mediated by interactions between biotic and abiotic factors. These differences in fungal community composition and individual taxa responses to host and environment interactions can explain niche partitioning effects on seedling survival [71] from the perspective of the fungal communities. Fungal taxa responding specifically to elevated temperature, both positive and negative, should be explored further to better understand the potential changes in plant-fungal interactions under climate change. The data presented here supports the hypothesis of mechanisms through which effective specialization allow generalist pathogens to mediate plant community dynamics [14].

## Supporting information

**S1 File. Supporting Figures S1A-S1J.**
(PDF)

**S2 File. Supporting Tables S2A-S2I.**
(PDF)

**S3 File. Supplementary Methods.**
(PDF)

**S4 File. Supporting Table S4.**
(XLSX)

## Acknowledgments

For field and lab assistance we thank B. Viera, B. Roper, B. Spakes-Richter, A. Barker Plotkin, A. Lewis, R. DeMatte, M. Campbell and M. Pasay. We thank the Environmental Metagenomics course at Sarah Lawrence College for their comments to this manuscript.

## Author contributions

**Conceptualization:** Michelle H. Hersh, Rytas Vilgalys, James S. Clark.

**Data curation:** Maria Soledad Benitez Ponce, Lindsey Becker.

**Formal analysis:** Maria Soledad Benitez Ponce.

**Funding acquisition:** Rytas Vilgalys, James S. Clark.

**Investigation:** Maria Soledad Benitez Ponce, Michelle H. Hersh, Lindsey Becker.

**Methodology:** Maria Soledad Benitez Ponce, Lindsey Becker.

**Project administration:** Rytas Vilgalys, James S. Clark.

**Resources:** Rytas Vilgalys, James S. Clark.

**Software:** James S. Clark.

**Supervision:** Rytas Vilgalys, James S. Clark.

**Visualization:** Maria Soledad Benitez Ponce, Michelle H. Hersh, Lindsey Becker.

**Writing – original draft:** Maria Soledad Benitez Ponce, Michelle H. Hersh, Lindsey Becker.

**Writing – review & editing:** Maria Soledad Benitez Ponce, Michelle H. Hersh, Lindsey Becker, Rytas Vilgalys, James S. Clark.

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
