## [Decision Letter · Decision Letter 0]

18 Nov 2024

PONE-D-24-34035Fungal community and taxa specialization to host and environment interactions in two temperate forestsPLOS ONE

Dear Dr. Benitez Ponce,

Thank you for submitting your manuscript to PLOS ONE. After careful consideration, we feel that it has merit but does not fully meet PLOS ONE’s publication criteria as it currently stands. Therefore, we invite you to submit a revised version of the manuscript that addresses the points raised during the review process.

Please carefully review the reviewers comments and make the necessary changes and respond to each of the comments provided by the reviewers. Pay particular attention to the comments of reviewer #2 on the methods section. Please see my comments in the methods section. In the current version the methods does not permit for evaluation of statistical power of the experiments or of the results. As suggested by reviewer #2, please list the number of samples that represented each statistical test so that the methods and statistical analyses are sufficient to be informative to the reviewers and in keeping with statistical analyses standards. 

We look forward to receiving your revised manuscript.

Kind regards,

Theodore Raymond Muth

Academic Editor

PLOS ONE

Journal Requirements:

3. Thank you for stating the following financial disclosure: [National Science Foundation grant NSF-DEB-0955904 to RV and JC].

4. Thank you for stating the following in the Acknowledgments Section of your manuscript: [For field and lab assistance we thank B. Viera, B. Roper, B. Spakes-Richter, A. Barker Plotkin, A. Lewis, R. DeMatte, M. Campbell and M. Pasay. We thank the Environmental Metagenomics course at Sarah Lawrence College for their comments to this manuscript. This work was funded by the National Science Foundation grant NSF-DEB-0955904.]

Please remove any funding-related text from the manuscript and let us know how you would like to update your Funding Statement. Currently, your Funding Statement reads as follows: [National Science Foundation grant NSF-DEB-0955904 to RV and JC].

5. Please include captions for your Supporting Information files at the end of your manuscript, and update any in-text citations to match accordingly. Please see our Supporting Information guidelines for more information: http://journals.plos.org/plosone/s/supporting-information .

6. We note that there is identifying data in the Supporting Information file <Benitez_etal_SuppMethods_7-24>. Due to the inclusion of these potentially identifying data, we have removed this file from your file inventory. Prior to sharing human research participant data, authors should consult with an ethics committee to ensure data are shared in accordance with participant consent and all applicable local laws.

-Location data

Additional guidance on preparing raw data for publication can be found in our Data Policy (https://journals.plos.org/plosone/s/data-availability#loc-human-research-participant-data-and-other-sensitive-data ) and in the following article: http://www.bmj.com/content/340/bmj.c181.long .

Reviewers' comments:

Reviewer's Responses to Questions

**Comments to the Author**

1. Is the manuscript technically sound, and do the data support the conclusions?

Reviewer #1: Yes

Reviewer #2: Partly

2. Has the statistical analysis been performed appropriately and rigorously?

Reviewer #1: Yes

Reviewer #2: No

3. Have the authors made all data underlying the findings in their manuscript fully available?

Reviewer #1: Yes

Reviewer #2: Yes

4. Is the manuscript presented in an intelligible fashion and written in standard English?

Reviewer #1: Yes

Reviewer #2: Yes

5. Review Comments to the Author

Reviewer #1: Review of the Manuscript Submitted to PLOS ONE by Ponce and Colleagues, Titled: “Fungal Community and Taxa Specialization to Host and Environment Interactions in Two Temperate Forests”

Summary

In this study, the authors tested the hypothesis that fungal host specialization occurs at both the individual fungus and fungal community levels and is influenced by environmental conditions. They evaluated the effects of various environmental factors on structuring seedling-associated fungal communities in temperate tree species, taking into account both the Janzen-Connell (JC) hypothesis and the impacts of climate warming.

To achieve their objective, the authors surveyed seedling-associated fungal communities from experimental sites in two temperate forests. The experimental design incorporated treatments involving light availability, elevated temperature, and distance to and density of conspecifics. The seedling mycobiome fungal communities were characterized using high-throughput sequencing, and the data were analyzed using generalized joint attribute models.

The results showed that host species, distance to conspecifics, and conspecific density were the strongest factors in shaping seedling fungal communities. The authors also identified specific fungal taxa that responded to these predictors. Overall, their findings support the concept of fungal host specialization at the community level and highlight particular fungal taxa that may contribute to density- and distance-dependent regulation of plant species diversity in the studied forests.

General Comments

The manuscript is well-written, with a high-quality experimental design, methods, and interpretation of results. I found it enjoyable to read. It makes a solid and significant contribution to our understanding of fungal-host-environment dynamics, which will be especially valuable for future models considering these dynamics in the context of global environmental changes. The results are likely to be of broad interest to the scientific community.

Specific Comments

My comments below are also available on the pdf copy of the manuscript.

1. Supplementary Materials and Methods (Page 5): Please add the similarity cutoff value used in “UPARSE algorithm implemented in USEARCH version 7 [10] was used for OTU calling at XX% similarity cutoff.”

2. Page 8, Line 165: I suggest including the expected size range for the amplicons generated with primers LR0R and LR3 of LSU. This information will help readers understand why short-read sequencing technology (e.g., Illumina) could have been used alongside long-read 454 pyrosequencing. Please also clarify the rationale for using both platforms.

3. Page 8, Line 168: Please provide the URL for Qiita.

4. Page 9, Line 182: Readers unfamiliar with the domain might not know what SILVA is. Please provide the URL for SILVA.

5. Table 1: The authors state, “Host species names in bold indicate hosts for which models of host and density and host and distance interaction were analyzed.” However, no species names are in bold. Please revise this accordingly.

6. Page 13, Lines 246–248: Please clarify what the numbers (e.g., Sordariomycetes_186) refer to.

7. Supplementary Figure S4: Please check the format and rendering of this figure. There seems to be an issue with the graph labels and lines. Additionally, the sequencing labels should distinguish between the two platforms used.

8. Table S8: Please add lines to the table for clarity.

9. Figure 3D: I suggest writing out "Duke Forest Eno West" for consistency, as "Harvard Forest" is written out in Figure 3C. Alternatively, use a complete acronym for "Harvard Forest ST."

10. Figure 4: The graph is difficult to interpret. It appears to show that more than 150 taxa had zero responses, which seems incorrect given that 247 minus 109 equals 138. Additionally, the tick marks should be placed in the middle of the bars to improve readability.

11. Page 19, Lines 382–388 (Page 20): This section reads like a figure legend, and the writing style differs from the preceding text. I suggest revising it to report specific observations and use past tense. The same applies to the next paragraph (Lines 390–399).

12. Discussion Section: I recommend removing references to figures and tables in the discussion, as they have already been addressed in the results section. Additionally, some sentences read like results and should be moved to the appropriate section. For example, “Most taxa responded to a limited number of predictors, but a small subset responded to more than ten (Fig. 4)” (Lines 425–426) should be placed in the results section.

Reviewer #2: Please see my comments in the methods section. Unclear and cryptic description of methods does not allow evaluation of statistical power of the experiments nor of the results. Please list the number of samples that represented each statistical test. There are a host of 'treatments' in the study and these should be listed clearly along with the 'n' that represented each test.

6. PLOS authors have the option to publish the peer review history of their article (what does this mean? ). If published, this will include your full peer review and any attached files.

**Do you want your identity to be public for this peer review?** For information about this choice, including consent withdrawal, please see our Privacy Policy .

Reviewer #1: **Yes: ** Martin Petrus Albertus Coetzee

Reviewer #2: No

---

## [Author Response · Author response to Decision Letter 1]

20 Jan 2025

The contents below are also included in the "Response to reviewers" file.

Manuscript title: Fungal community and taxa specialization to host and environment interactions in two temperate forests

The authors thank the editor and reviewers for their thorough assessment of this manuscript. The manuscript has been revised per your recommendations, and in the text below specific aspects of the revisions are indicated per comment. Reviewer comments are indicated in regular font, and responses in italics. A version of the manuscript including track changes is submitted along with a clean version as a reference. Line numbers referred to in this text correspond with the “tracked changes on” version of the manuscript.

Editor comments:

Please carefully review the reviewers comments and make the necessary changes and respond to each of the comments provided by the reviewers. Pay particular attention to the comments of reviewer #2 on the methods section. Please see my comments in the methods section. In the current version the methods does not permit for evaluation of statistical power of the experiments or of the results. As suggested by reviewer #2, please list the number of samples that represented each statistical test so that the methods and statistical analyses are sufficient to be informative to the reviewers and in keeping with statistical analyses standards.

Thank you for your suggestions. We have included additional information regarding number of samples in this study in the main manuscript text, as well as the Supplementary Methods File (S3 File) and the Supporting Table S2F in S2 File.

We have also thoroughly revised the manuscript to incorporate journal requirements for style, file naming, permits, supporting information captions. We also removed the financial disclosure information from the acknowledgements. We noticed that one of the Supporting Files was removed from the original submission (Supplementary Methods File, now file S3). This file has additional information on the sampling locations and methods.

Reviewers' comments:

Reviewer #1: Review of the Manuscript Submitted to PLOS ONE by Ponce and Colleagues,

Titled: “Fungal Community and Taxa Specialization to Host and Environment Interactions in

Two Temperate Forests”

The manuscript is well-written, with a high-quality experimental design, methods, and

interpretation of results. I found it enjoyable to read. It makes a solid and significant contribution to our understanding of fungal-host-environment dynamics, which will be especially valuable for future models considering these dynamics in the context of global environmental changes. The results are likely to be of broad interest to the scientific community.

We thank the reviewer for the positive feedback.

Specific Comments

My comments below are also available on the pdf copy of the manuscript.

1. Supplementary Materials and Methods (Page 5): Please add the similarity cutoff value used in “UPARSE algorithm implemented in USEARCH version 7 [10] was used for OTU calling at XX% similarity cutoff.”

Done

2. Page 8, Line 165: I suggest including the expected size range for the amplicons generated with primers LR0R and LR3 of LSU. This information will help readers understand why short-read sequencing technology (e.g., Illumina) could have been used alongside long-read 454 pyrosequencing. Please also clarify the rationale for using both platforms.

Done (Ln 180 and 182-183).

3. Page 8, Line 168: Please provide the URL for Qiita.

Done (Ln 185)

4. Page 9, Line 182: Readers unfamiliar with the domain might not know what SILVA is. Please provide the URL for SILVA.

Done (Ln 201-202)

5. Table 1: The authors state, “Host species names in bold indicate hosts for which models of host and density and host and distance interaction were analyzed.” However, no species names are in bold. Please revise this accordingly.

The table legend was revised. All host species in the current version of the manuscript were included in all models.

6. Page 13, Lines 246–248: Please clarify what the numbers (e.g., Sordariomycetes_186) refer to in the manuscript.

Depending on the OTU we were able to classify taxa at the genus level or above. The naming convention used in this manuscript for each individual OTU indicates the lowest classification category followed by a number to differentiate between OTUs within the same taxonomic level (e.g. Colletotrichum_1 and Colletotrichum_2 correspond to two different OTUs of Colletotrichum; whereas for Sordariomycetes, if considering the full dataset of 2975 OTUs, we had at least 186 OTUs that were classified with confidence only to the class Sordariomycetes). A similar statement was included in the Methods section (Ln 205-207).

7. Supplementary Figure S4: Please check the format and rendering of this figure. There seems to be an issue with the graph labels and lines. Additionally, the sequencing labels should distinguish between the two platforms used.

Done

8. Table S8: Please add lines to the table for clarity.

Done

9. Figure 3D: I suggest writing out "Duke Forest Eno West" for consistency, as "Harvard Forest" is written out in Figure 3C. Alternatively, use a complete acronym for "Harvard Forest ST."

Done, changed to Duke Forest EW to keep consistency in both panels.

10. Figure 4: The graph is difficult to interpret. It appears to show that more than 150 taxa had zero responses, which seems incorrect given that 247 minus 109 equals 138. Additionally, the tick marks should be placed in the middle of the bars to improve readability.

Thank you for noticing this. The graph has been redrawn.

11. Page 19, Lines 382–388 (Page 20): This section reads like a figure legend, and the writing

style differs from the preceding text. I suggest revising it to report specific observations and use past tense. The same applies to the next paragraph (Lines 390–399).

Done.

12. Discussion Section: I recommend removing references to figures and tables in the discussion, as they have already been addressed in the results section. Additionally, some sentences read like results and should be moved to the appropriate section. For example, “Most taxa responded to a 7 of 7 limited number of predictors, but a small subset responded to more than ten (Fig. 4)” (Lines 425– 426) should be placed in the results section.

References to figures and tables were removed from the discussion section. Regarding the statements about results in the discussion section, these were kept to help provide context or examples in the discussion section.

Reviewer #2: Please see my comments in the methods section. Unclear and cryptic description of methods does not allow evaluation of statistical power of the experiments nor of the results. Please list the number of samples that represented each statistical test. There are a host of 'treatments' in the study and these should be listed clearly along with the 'n' that represented each test.

Thank you for your comments. The manuscript has been revised in different sections to include information of sample sizes. Examples of these could be found in the Methods section, S3 File Supporting Methods and Supporting Table S2F in S2 File.

---

## [Decision Letter · Decision Letter 1]

23 Mar 2025

Fungal community and taxa specialization to host and environment interactions in two temperate forests

PONE-D-24-34035R1

Dear Dr. Benitez Ponce,

We’re pleased to inform you that your manuscript has been judged scientifically suitable for publication and will be formally accepted for publication once it meets all outstanding technical requirements. It was noted by reviewers that the manuscripts lacks NCBI accession numbers and other raw data links. Please address this.

Kind regards,

Theodore Raymond Muth

Academic Editor

PLOS ONE

Additional Editor Comments (optional):

Reviewers' comments:

Reviewer's Responses to Questions

**Comments to the Author**

1. If the authors have adequately addressed your comments raised in a previous round of review and you feel that this manuscript is now acceptable for publication, you may indicate that here to bypass the “Comments to the Author” section, enter your conflict of interest statement in the “Confidential to Editor” section, and submit your "Accept" recommendation.

Reviewer #1: All comments have been addressed

Reviewer #2: All comments have been addressed

2. Is the manuscript technically sound, and do the data support the conclusions?

Reviewer #1: Yes

Reviewer #2: Yes

3. Has the statistical analysis been performed appropriately and rigorously?

Reviewer #1: Yes

Reviewer #2: Yes

4. Have the authors made all data underlying the findings in their manuscript fully available?

Reviewer #1: Yes

Reviewer #2: No

5. Is the manuscript presented in an intelligible fashion and written in standard English?

Reviewer #1: Yes

Reviewer #2: Yes

6. Review Comments to the Author

Reviewer #1: The authors have adequately addressed my comments in my previous review. I, therefore, feel that this manuscript is now acceptable for publication pending the comments from the second reviewer (if applicable).

Reviewer #2: (No Response)

7. PLOS authors have the option to publish the peer review history of their article (what does this mean? ). If published, this will include your full peer review and any attached files.

**Do you want your identity to be public for this peer review?** For information about this choice, including consent withdrawal, please see our Privacy Policy .

Reviewer #1: **Yes: ** Martin Petrus Albertus Coetzee

Reviewer #2: No

---

## [Editor Report · Acceptance letter]

PONE-D-24-34035R1

PLOS ONE

Dear Dr. Benitez Ponce,

I'm pleased to inform you that your manuscript has been deemed suitable for publication in PLOS ONE. Congratulations! Your manuscript is now being handed over to our production team.

Kind regards,

on behalf of

Dr. Theodore Raymond Muth

Academic Editor

PLOS ONE